# LEARNING FROM ONE AND ONLY ONE SHOT

## ABSTRACT

Humans can generalize from only a few examples and from little pre-training on similar tasks. Yet, machine learning (ML) typically requires large data to learn or pre-learn to transfer. Inspired by nativism, we directly model basic human-innate priors in abstract visual tasks e.g., character/doodle recognition. This yields a white-box model that learns transformation-based topological similarity by mimicking how humans naturally "distort" an object when first seeing it. Using the simple nearest-neighbor classifier in this similarity space, our model achieves human-level character recognition using only 1–10 examples per class and nothing else (no pre-training). This differs from few-shot learning (FSL) using significant pre-training. On standard benchmarks MNIST, EMNIST-letters, and the Omniglot challenge, our model outperforms both neural-network-based and classical ML in the "tiny-data" regime, including FSL pre-trained on large data. Further, mimicking $k$-means but in a non-Euclidean space, our model enables unsupervised learning and generates human-interpretable archetypes as cluster "centroids".

## 1 INTRODUCTION

Modern machine learning (ML) has made remarkable progress, but this is accompanied by increasing model complexity, with hundreds of neural layers (e.g., ResNet-152) and millions of parameters (e.g., AlexNet: 62.3M, VGG16: 138M, BERT: 110M, GTP-3: 175B). This results in a huge appetite for data and increasing difficulty in model interpretability—both for users to understand and for developers to tune (e.g., hyperparameters, architecture). As such, AI researchers have pushed for ML models that are *prior-* and *data-efficient* (Chollet, 2019), that are *human-like* (Lake et al., 2015), and that exhibit *human-interpretable* behaviors (Adadi & Berrada, 2018).

This poses the fundamental scientific question: How can humans learn so much from so little (May, 2015), whereas ML models, e.g., those achieving near-perfection on MNIST using all 60k training images, deteriorate rapidly as (pre)training reduces? Being data-hungry can pose a real challenge in data-scarce domains, e.g., in a rapidly evolving pandemic or a low-resource environment. Focusing on its *scientific contribution*, this paper presents a theoretically sound, white-box model that learns like humans do, with initial success on a first set of benchmarks. This includes our state-of-the-art results of hitting $80\%$ / $90\%$ MNIST accuracy using the only first / first four training images per class and achieving $6.75\%$ error (human performance is $4.5\%$) in the Omniglot one-shot learning challenge without pre-training—one and only one shot.

We follow the nativist principle: given an example, humans make *abstractions*: we envision an equivalence class of unseen examples, equivalent to the given, in several abstract senses. Many abstraction abilities are inborn in humans and occur unconsciously. When a baby sees a mug, (s)he can immediately recognize it regardless of whether it is translated, rotated, scaled, or deformed. Such abstraction abilities are considered innate as Core Knowledge priors (Spelke & Kinzler, 2007), rather than acquired later in life such as learning that a mug is topologically a doughnut.

We computationally realize the above intuition via our so-called *distortable canvas*—imagining every image smoothly painted on a rubber canvas that can be distorted in many ways, e.g., bent, stretched, squeezed (Figure 1A). Due to rubber's viscosity, more distorted canvas transformations expend more energy. This induces a topological similarity, or distance, based on *minimal energy*: two images are similar if one can almost transform into the other with little distortion (energy). Distance is computed by minimizing color and canvas distortions. In general, visual similarity involves not only a canvas but also a color transformation. Here, we focus on canvas transformations and grayscale images only.

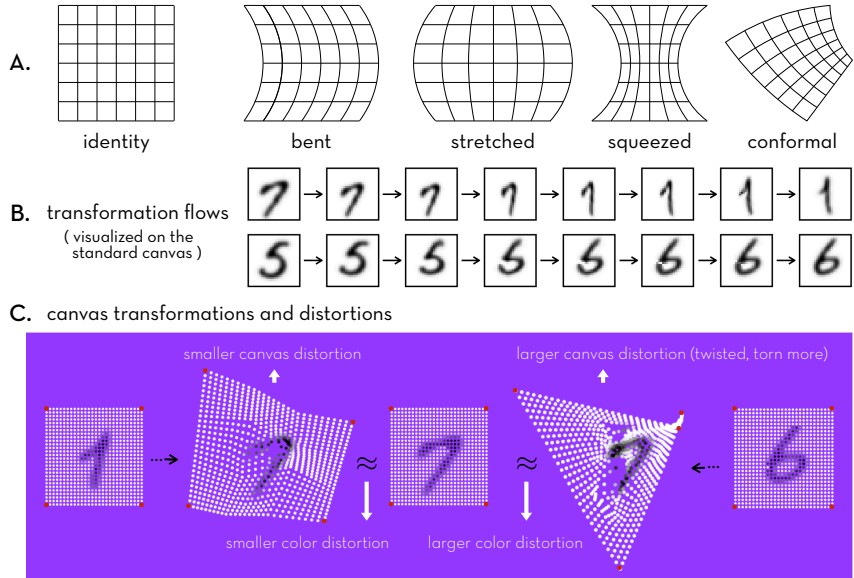

Figure 1: Canvas transformations (A), transformation flows (B), and distortions (C).

Besides a final desired transformation (and distance), we apply the minimal-energy principle to the entire transformation process, i.e., to keep canvas and color distortions small along the entire optimization process (Mesa et al., 2019). When perceiving a translation, our mind does not process it as a sudden displacement from one location to another, but auto-completes a translation path—continuous and preferably short. Gradient descent naturally fits this goal by always choosing the steepest descent. Yet, it suffers from the *curse of local minima*. Our solution is to *lift gradient descent to multiple levels of abstraction* via multiscale canvas lattices and color blurring, mimicking human abstraction ability that is extremely flexible in multiscale optimization. This yields visualizable and interpretable transformation flows (Figure 1B) that either match human intuition (e.g., what humans would naturally do to transform a "7" to a "1") or provide human intuition to initially nonintuitive settings. The latter case suggests new, human-interpretable transformations: "ah, I did not realize this other way of transforming '7' into '1', but now I see it and it makes perfect sense!"

We show initial success on abstract visual tasks such as character and doodle recognition (future work generalizes to photorealistic images after a preprocessing technique to turn them into abstract doodles or "emojis"). Our model can be used with the simple nearest-neighbor method to classify images and also in a simple $k$-means style to cluster images. On benchmarks including MNIST (LeCun et al., 1998), EMNIST-letters (Cohen et al., 2017), and the more taxing Omniglot challenge (Lake et al., 2015), running nearest-neighbor on our learned similarity space outperforms both neural-network-based and classical ML in the "tiny-data" or single-datum regime. This includes beating few-shot models pre-trained on extra background data (which our model does not require). On MNIST, we need one and only one training image per class to hit $80\%$ accuracy and four to hit $90\%$. In an unsupervised setting, our model enables $k$-means-like clustering, but on our learned similarity space. We generate archetypes as cluster "centroids", e.g., different ways of writing "7" or doodling giraffes.

**Literature on data scarcity: few-shot learning (FSL).** One-/few-shot learning (Lake et al., 2011; Wang et al., 2020) via transfer or meta learning (Pan & Yang, 2009; Finn et al., 2017; Hsu et al., 2019) has achieved impressive success in data-scarce scenarios. FSL's success relies on the assumption that source and target tasks are similar enough for pre-training to be relevant (Storkey, 2009). However, knowing *a priori* how relevant the tasks are and understanding what has been pre-learned are often considered "black art". It remains challenging for FSL practitioners to pick proper source data/models for pre-training so as to transfer most effectively and avoid *negative transfer* (Pan & Yang, 2009; Meiseles & Rokach, 2020). This is especially the case in new domains (e.g., discovering rising trends) over classical ones (e.g., vision, language). So, rather than "big transfer" (Kolesnikov et al., 2020), the Omniglot challenge urges "small transfer"—advocating reduced pre-training (Lake et al., 2019). This paper follows Omniglot's pursuit and pushes such reduction to the limit: with absolutely zero pre-training, we still achieved near-human performance and human-interpretability.

**Literature on data scarcity: transformation and data augmentation.** Our distortable canvas is conceptually akin to other transformation-based models, e.g., optimal-transport maps (Villani, 2009), transformation-induced descriptors (Lowe, 1999) and equivariances (Bronstein et al., 2017). Like all these models, we build transformations into the model rather than into the data as through data augmentation (Krizhevsky et al., 2012). This has the native advantage of harnessing transformational properties directly rather than learning them from data. Models with built-in transformations may be viewed as those perfectly learned from infinitely augmented data. Unlike transformations commonly considered in existing models and data augmentation techniques, we do not encode domain knowledge about preferred transformations such as translation, rotation, or scaling. Instead, our model considers all transformations while still maintaining efficiency via our abstracted gradient descent.

## 2 SMOOTH IMAGE ON DISTORTABLE CANVAS

We introduce a *distortable canvas* model: any image is thought of as smoothly painted on a rubber canvas that can be bent, stretched, etc. We further introduce *canvas transformations* that can flexibly "distort" an image as we naturally simulate in our mind. More specifically, we define a *smooth image* by a piecewise differentiable $\mathcal{M} : \mathbb{R}^2 \to \mathbb{R}_+$, where $\mathbb{R}^2$ denotes an infinite *canvas* and $\mathbb{R}_+$ denotes *color* (grayscale in this paper). We define a *canvas transformation* by $\alpha : \mathbb{R}^2 \to \mathbb{R}^2$, which "reshapes" the underlying canvas of a smooth image. Examples include translation, rotation, scaling, and more. We also define a *color transformation* by $\chi : \mathbb{R}_+ \to \mathbb{R}_+$, which "repaints" a color. In this paper, we simplify color transformation and only use it to adjust image contrast via affine $\chi(c) := ac + b$. Nevertheless, we do *not* restrict canvas transformation, but consider *all* 2D transformations. Given $\mathcal{M}, \alpha, \chi$, the composition $\chi \circ \mathcal{M} \circ \alpha$ denotes the transformed image of $\mathcal{M}$ by transformations $\alpha, \chi$.

To mimic innate human intuition about topological similarity, we introduce *canvas distortion* $\mathcal{D}_V(\alpha)$ for any canvas transformation $\alpha$ and *color distortion* $\mathcal{D}_C(\mathcal{M}, \mathcal{M}')$ between two smooth images $\mathcal{M}, \mathcal{M}'$. Our idea is to search for a transformation that mimics what humans naturally do to transform one image into another. That is, a low-distorted $\alpha$ which makes little difference in color between $\mathcal{M}$ and the transformed $\mathcal{M}'$, or more precisely, an $\alpha$ that minimizes both $\mathcal{D}_V(\alpha)$ and $\mathcal{D}_C(\mathcal{M}, \chi \circ \mathcal{M}' \circ \alpha)$.

**Representating digital and smoothed images.** An $m \times n$ *digital image* is a discrete $\mathsf{M} : [m] \times [n] \to [0, 1]$, where $[k] := \{0, 1, \ldots, k-1\}$. We call $[m] \times [n]$ the *canvas grid* and any $z \in [m] \times [n]$ a *grid point*. For any $m \times n$ digital image $\mathsf{M}$, we smooth it to $\mathcal{M}$ via a *sum of kernels*:

$$\mathcal{M}(x) := \sum_{z \in [m] \times [n]} \mathsf{M}(z) \cdot \kappa(\rho(z, x)) \quad \text{for any } x \in \mathbb{R}^2, \tag{1}$$

where a kernel $\kappa : \mathbb{R}_+ \to \mathbb{R}_+$ is a decaying function (e.g., linear, polynomial, Gaussian decay) and $\rho$ is a metric on $\mathbb{R}^2$ (e.g., $\ell_1, \ell_2, \ell_\infty$). In this paper, we use linear decay and $\ell_\infty$, i.e., $\kappa(\rho(z, x)) = 1 - \frac{1}{\rho_c} \|z - x\|_\infty$ if $\|z - x\|_\infty < \rho_c$ (for some cutoff radius $\rho_c > 0$) and $\kappa(\rho(z, x)) = 0$ otherwise. Note: $\mathcal{M}$ is defined everywhere on $\mathbb{R}^2$. This differs from Gaussian blurring as we do not discretize kernels. It is key to use the smoothed image as input, which allows computing gradients analytically. As such, we always smooth any digital image at first and then only manipulate the smoothed image.

**Representing arbitrary canvas transformations.** We consider all 2D transformations (including those without a formula), but how do we represent them in a computer? With respect to the *standard grid* $[m] \times [n]$, we use the *transformed grid* $\alpha([m] \times [n])$ to represent $\alpha$ digitally. Thus, any canvas transformation $\alpha$ is *digitally represented by* ($\overset{d}{=}$) a matrix $\boldsymbol{\alpha} \in \mathbb{R}^{(mn) \times 2}$ whose $i$th row is the 2D coordinate of the transformed $i$th grid point. We use the lexicographical order of a 2D grid, e.g., with respect to $[2] \times [3]$, the identify transformation id $\overset{d}{=}$ **id** $= [[0, 0], [0, 1], [0, 2], [1, 0], [1, 1], [1, 2]]$. Any transformed image $\mathcal{M} \circ \alpha \overset{d}{=} \mathcal{M}(\boldsymbol{\alpha}) := (\mathcal{M}(\boldsymbol{\alpha}_0), \ldots, \mathcal{M}(\boldsymbol{\alpha}_{(mn-1)})) \in \mathbb{R}^{(mn)}$, i.e., a (vectorized) digital image sampled from $\mathcal{M}$ at the transformed grid $\boldsymbol{\alpha}$.

**Representing color and canvas distortions.** The color distortion $\mathcal{D}_C$ measures the color discrepancy between $\mathcal{M}(\mathbf{id})$ and $\mathcal{M}'(\boldsymbol{\alpha})$ up to an affine color transformation $\chi$. The canvas distortion $\mathcal{D}_V$ measures the distortion between the original grid **id** and the transformed grid $\boldsymbol{\alpha}$. Formally,

$$\mathcal{D}_C(\mathcal{M}, \chi \circ \mathcal{M}' \circ \alpha) \overset{d}{=} \mathcal{D}_C(\mathcal{M}(\mathbf{id}), \chi(\mathcal{M}'(\boldsymbol{\alpha}))) := \|a\mathcal{M}'(\boldsymbol{\alpha}) + b - \mathcal{M}(\mathbf{id})\|_2^2 \tag{2}$$

$$\mathcal{D}_V(\alpha) \overset{d}{=} \mathcal{D}_V(\mathbf{id}, \boldsymbol{\alpha}) := \max_{\{\{i,j\}, \{i',j'\}\} \in B_E} \left| \Delta_{\{i,j\}}^{\boldsymbol{\alpha}} - \Delta_{\{i',j'\}}^{\boldsymbol{\alpha}} \right|, \quad \Delta_{\{i,j\}}^{\boldsymbol{\alpha}} := \log \frac{\|\boldsymbol{\alpha}_i - \boldsymbol{\alpha}_j\|_2}{\|\mathbf{id}_i - \mathbf{id}_j\|_2} \tag{3}$$

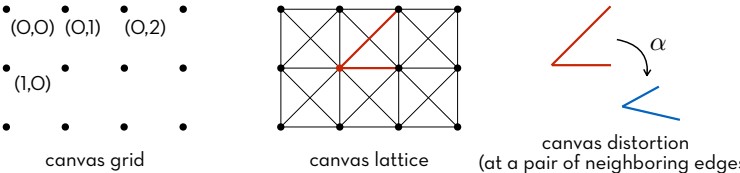

Figure 2: Canvas grid and its corresponding lattice. Local distortions are computed at every pair of neighboring edges. One example of neighboring edges is highlighted in red.

Here, $B_E$ comprises all pairs of neighboring edges in a *canvas lattice* (introduced below). Eq. (3) is derived from the mathematical definition of *distortion of a function* by discretizing it across the canvas lattice. This formula measures how far an arbitrary transformation is from being *conformal*, which is flexible for *local* isometries and scaling. Given a canvas grid $[m] \times [n]$, its corresponding *canvas lattice* is an undirected graph $L = (V, E)$, with the set of vertices $V = [m] \times [n]$ and the set of edges obtained by connecting neighboring vertices in the $\ell_\infty$ sense: $E = \{\{i, j\} \mid \|v_i - v_j\|_\infty = 1 \text{ for } v_i, v_j \in V\}$. We say two edges are *neighbors* if they form a $45°$ angle (Figure 2).

**Computing topological distance by minimizing distortions.** To minimize color and canvas distortions (2) and (3), we consider two dual views: minimizing $\mathcal{D}_C$ among low-distorted $\alpha$'s or minimizing $\mathcal{D}_V$ among best-matching $\alpha$'s. We write the two views as the following two constrained optimization problems, together with their respective unconstrained equivalents: with $\epsilon \to 0_+$ and $\mu \to 0_+$,

$$\min_{\alpha,\chi} \mathcal{D}_C(\mathcal{M}, \chi \circ \mathcal{M}' \circ \alpha) \quad \text{s.t. } \mathcal{D}_V(\alpha) \leq \epsilon \iff \min_{\alpha,\chi} \mathcal{D}_V(\alpha) + \mu \mathcal{D}_C(\mathcal{M}, \chi \circ \mathcal{M}' \circ \alpha) \quad (4)$$

$$\min_{\alpha,\chi} \mathcal{D}_V(\alpha) \quad \text{s.t. } \mathcal{D}_C(\mathcal{M}, \chi \circ \mathcal{M}' \circ \alpha) \leq \epsilon \iff \min_{\alpha,\chi} \mathcal{D}_C(\mathcal{M}, \chi \circ \mathcal{M}' \circ \alpha) + \mu \mathcal{D}_V(\alpha) \quad (5)$$

We let the optima $\mathcal{D}_C^\star$ for (4) and $\mathcal{D}_V^\star$ for (5) denote two versions of our desired topological distance that mimics innate human intuition. We call them $\mathcal{D}_C$-*distance* and $\mathcal{D}_V$-*distance*, respectively.

**Transformation flow.** To obtain both transformations and transformation processes that are human-like, we run (projected) gradient descent. The iterative gradient steps yield not only a transformation $\alpha^\star$ in the end but also a *transformation flow* $\text{id} = \alpha^{(0)} \to \alpha^{(1)} \to \cdots \to \alpha^\star$. The resulting sequence of transformed images $\mathcal{M}' = \mathcal{M}' \circ \alpha^{(0)} \to \mathcal{M}' \circ \alpha^{(1)} \to \cdots \to \mathcal{M}' \circ \alpha^\star \approx \mathcal{M}$ (we omit $\chi$ for simplicity) makes up an animation (Figure 1), which helps with human intuition on transforming $\mathcal{M}'$ to $\mathcal{M}$. However, directly running (projected) gradient descent on (4) or (5) does not work, because it suffers from the curse of local minima, which we discuss and solve in the next section.

## 3 GRADIENT DESCENT AT HIGHER LEVELS OF ABSTRACTION

The canvas distortion $\mathcal{D}_V$ is invariant under a variety of transformations (e.g., $\mathcal{D}_V(\alpha) = 0$ for any conformal $\alpha$), which nicely mimics humans' flexible transformation options. But this also implies lots of local/global minima and other critical points where the gradient is zero. How much the color distortion $\mathcal{D}_C$ fluctuates as a function of $\alpha$ depends on the images $\mathcal{M}, \mathcal{M}'$. But in most cases, $\mathcal{D}_C$ also has lots of local/global minima, the majority of which represent unwanted "short cuts"—unnatural transformations that make $\mathcal{D}_C \to 0$ but would break the rubber canvas or create holes in it. The curse of vanishing gradients can freeze gradient descent. To unfreeze it, we lift gradient descent to higher levels, mimicking once again humans' abstraction power, as our internal optimization system is quite flexible in pursuing "gradient-descent" moves at multiple levels of abstraction. We design two abstraction techniques: a chain of anchor lattices to make hierarchical abstractions of canvas transformations and a chain of color blurring to make hierarchical abstractions of image painting.

**Anchor grids and lattices.** An *anchor grid* and its corresponding *anchor lattice* offer a simpler parameterization (i.e., an abstraction) of canvas transformations. Without such an abstraction, any transformed $[m] \times [n]$ grid $\boldsymbol{\alpha} \in \mathbb{R}^{(mn) \times 2}$ consists of $2mn$ free parameters. So, the optimization problems (4) and (5) are $2mn + 2$ dimensional, which is not only computationally inefficient for large images but also has too much room for vanishing gradient. We use a simpler $\boldsymbol{\alpha}$-parameterization that regularizes transformation, lowers distortion, and agrees with our intuition on rubber transformations.

Formally, an *anchor system* $(G, \hat{G}) = (M \times N, \hat{M} \times \hat{N})$ uses two layers of grids: an *underlying grid* $G$ and an *anchor grid* $\hat{G}$ atop, satisfying $\hat{M} \subseteq M, \hat{N} \subseteq N$, and $G \subseteq \text{ConvexHull}(\hat{G})$. Figure 3A shows

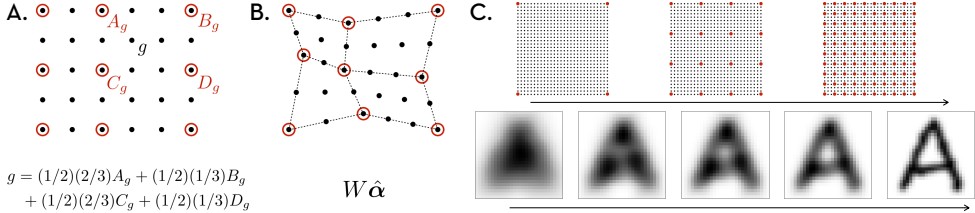

$g = (1/2)(2/3)A_g + (1/2)(1/3)B_g$
$+ (1/2)(2/3)C_g + (1/2)(1/3)D_g$

$W\hat{\alpha}$

Figure 3: An anchor system (A) and its transformation (B). (C) exemplifies a configuration of $(\hat{G}, \rho_c)$-solution path consisting of a chain of anchor grids/lattices and a chain of blurring.

one example, where $G = [5] \times [6] = \{0, \ldots, 4\} \times \{0, \ldots, 5\}$ and $\hat{G} = \{0, 2, 4\} \times \{0, 2, 5\}$. Under an anchor system, we can uniquely represent any grid point $g \in G$ via four anchors $A_g, B_g, C_g, D_g \in \hat{G}$ via proportional interpolation, or more precisely, the following double convex combination

$$g = (1 - \lambda_g)(1 - \nu_g)A_g + (1 - \lambda_g)\nu_g B_g + \lambda_g(1 - \nu_g)C_g + \lambda_g \nu_g D_g. \tag{6}$$

Here, $A_g B_g D_g C_g$ can be uniquely selected as the smallest rectangle in $\hat{G}$'s lattice containing $g$; the two weight parameters $\lambda_g, \nu_g$ are computed based on relative position, e.g., as in Figure 3A. The relation between grid points and anchors can be summarized by a weight matrix $W \in \mathbb{R}^{|G| \times |\hat{G}|}$. Its $i$th row stores weights for the $i$th grid point (say $g$ in (6)) and contains at most four non-zero entries (i.e., coefficients in (6)) located at the columns corresponding to $A_g, B_g, C_g, D_g$, respectively.

Given an anchor system $(G, \hat{G})$, any canvas transformation $\alpha \stackrel{d}{=} \boldsymbol{\alpha} \in \mathbb{R}^{|G| \times 2}$ under $G$ and $\stackrel{d}{=} \hat{\boldsymbol{\alpha}} \in \mathbb{R}^{|\hat{G}| \times 2}$ under $\hat{G}$. $\hat{\boldsymbol{\alpha}}$ is a submatrix of $\boldsymbol{\alpha}$, which induces an equivalence relation on the set of all canvas transformations: $\boldsymbol{\alpha}, \boldsymbol{\beta}$ are equivalent iff $\hat{\boldsymbol{\alpha}} = \hat{\boldsymbol{\beta}}$, and $\hat{\boldsymbol{\alpha}}$ abstracts the equivalence class $\{\boldsymbol{\beta} \mid \hat{\boldsymbol{\beta}} = \hat{\boldsymbol{\alpha}}\}$. Based on the maximum entropy principle (Jaynes, 1957), a reasonable selection of a representative of this equivalence class is $W\hat{\boldsymbol{\alpha}}$, because $W\hat{\boldsymbol{\alpha}} \in \{\boldsymbol{\beta} \mid \hat{\boldsymbol{\beta}} = \hat{\boldsymbol{\alpha}}\}$ and evenly distributes the transformed grid points. Figure 3B illustrates this type of even distribution, which agrees with human intuition on how a rubber surface would naturally react when transforming forces are applied at anchors.

Using an anchor system in optimization problems (4) and (5) adds very little to computing distortions and gradients: we reuse the computation with $\boldsymbol{\alpha} = W\hat{\boldsymbol{\alpha}}$ and perform only one additional chain-rule step $\partial \boldsymbol{\alpha} / \partial \hat{\boldsymbol{\alpha}} = W$. By doing so, however, the number of optimization variables in (4) or (5) reduces from $|G| + 2$ to $|\hat{G}| + 2$ (e.g., if $G = [28] \times [28]$ and $\hat{G} = \{0, 27\} \times \{0, 27\}$, the number reduces from 1570 to 10). It is important to note that using a simpler anchor grid is *not* the same as downsampling. If it were, one would plug in $\alpha \leftarrow \hat{\boldsymbol{\alpha}}$, but we plug in $\alpha \leftarrow W\hat{\boldsymbol{\alpha}}$. In our case, image colors are still sampled from the underlying grid rather than downsampled from the anchor grid. So, using our anchor system is not information lossy while still benefiting from reduced optimization size. Running gradient descent (w.r.t. anchors) in abstracted optimization spaces effectively bypasses critical points.

**Blurring.** Another view to lifting gradient descent to a high-level, abstracted optimization space, is to blur the image. Intuitively, blurring ignores low-level fluctuation, similar to how humans naturally abstract an image. Blurring helps remedy vanishing gradients and is done in our image smoothing process. The cutoff radius $\rho_c$ in $\kappa$ in (1) controls the blurring extent: larger $\rho_c$ means more blurred.

**Guided gradient descent.** Mixing the two abstraction techniques yields our guided gradient descent proceeding from higher- to lower-level abstractions. Given an anchor grid $\hat{G}$ and a cutoff radius $\rho_c$, we denote the corresponding (4) and (5) by $DC(\hat{G}, \rho_c)$ and $DV(\hat{G}, \rho_c)$, respectively. For either, we solve for a $(\hat{G}, \rho_c)$-solution path, from coarser $\hat{G}$ and larger $\rho_c$ to finer $\hat{G}$ and smaller $\rho_c$. Let $\hat{G}_k$ be a $k \times k$ evenly distributed anchor grid and $\hat{L}_k$ be its corresponding lattice. Figure 3C shows a chain of anchor lattices $\{\hat{L}_{3^i+1}\}_{i=0,1,2,\ldots}$ and cutoff radii $\{\eta^j \rho_{c_0}\}_{j=0,1,2,\ldots}$. It is easy initially to align two blurred blobs via small canvas adjustments, implying a small number of iterations to converge to $\mathcal{D}_C \approx \mathcal{D}_V \approx 0$. As we proceed along the solution path, the images restore more detail but the finer $\hat{L}_k$ helps manage that detail. In a solution path, an earlier solution is used to *warm start* the subsequent solve step, which further alleviates the curse of vanishing gradients. Notably, even the starting $\hat{L}_2$ comprising only four corner anchors parameterizes a large family of transformations containing all affine transformations. Finer anchor grids/lattices express more flexible transformations (including local, global, piecewise affine, and more), approaching human-level flexibility.

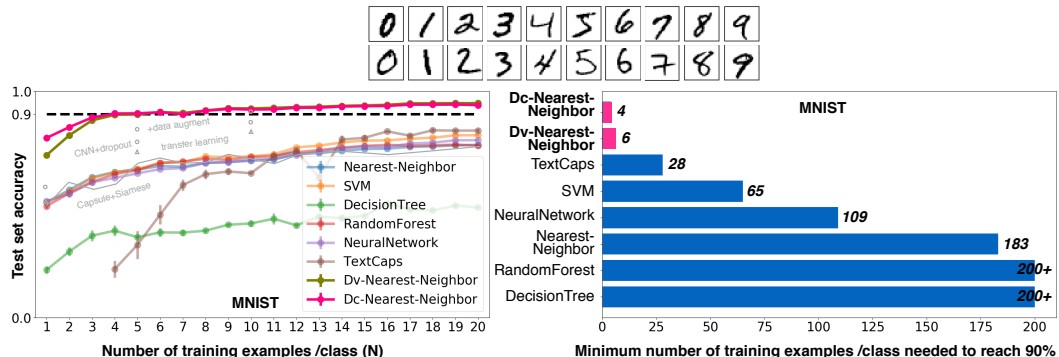

Figure 4: MNIST (10 classes) in the tiny-data regime: first 1–20 training images per class and full test set (examples shown on top). For each model listed in the legend, we plot its test accuracy versus the training size $N$ (bottom left) and also the smallest $N$ needed to reach a threshold of $90\%$ accuracy (bottom right). Our model outperforms all other models for all $N \in \{1, \dots, 20\}$, requiring the fewest training examples (first four or six per class) to reach $90\%$ accuracy.

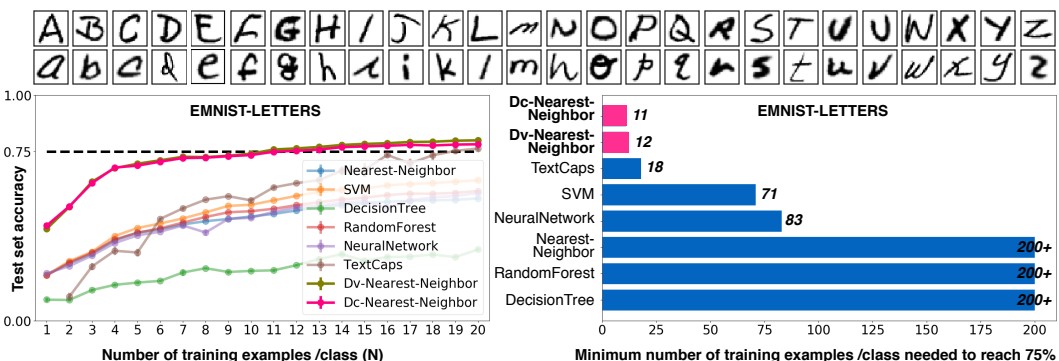

Figure 5: EMNIST-letters (26 classes) in the tiny-data regime: first 1–20 training images per class and full test set (examples shown on top). Results are shown in the same way as in Figure 4. Our model outperforms all other models for all $N \in \{1, \dots, 20\}$, requiring the fewest training examples (first eleven or twelve per class) to reach $75\%$ accuracy (due to increased difficulty in this dataset).

## 4 IMAGE CLASSIFICATION IN THE TINY DATA REGIME

We use our learned $\mathcal{D}_C$- / $\mathcal{D}_V$-distance in the simplest nearest-neighbor method to classify grayscale images, named $\mathcal{D}_C$- / $\mathcal{D}_V$-nearest-neighbor. The whole process of metric learning and classification is human intuitive and interpretable. We show classification performances on three standard benchmarks: the MNIST and EMNIST datasets of handwritten digits and letters restricted to the tiny-data regime, as well as the Omniglot challenge.

**MNIST in the tiny-data regime.** The original benchmark has 60k images for training and 10k for testing, spanning 10 classes. To evaluate how a model performs in the tiny data regime, we train the model on the first $N$ images per class from the original training set, test it on the full test set, and record test accuracy versus $N = 1, 2, 3, \dots$. We compare our model to both neural-network-based and classical ML models, including TextCaps (Jayasundara et al., 2019) with state-of-the-art performance in the small-data regime, SVM, nearest neighbor, etc. Classical ML is included to show that nailing the tiny-data regime does not mean just using simple models. For stochastic models, we record mean and standard deviation from 5 independent runs. TextCaps only runs when $N \geq 4$ and sometimes returns a random guess ($10\%$), so, we record trimmed mean and standard deviation from 11 runs (where we trim the best two and worst four). We also copy results from other references that ran MNIST in a similar tiny-data setting, including FSL that uses extra data for pre-training (whereas all our other selected models do not). These results are from the same training-testing sizes but not the same data sets, and hence, are considered indirect comparisons. We present all results in Figure 4.

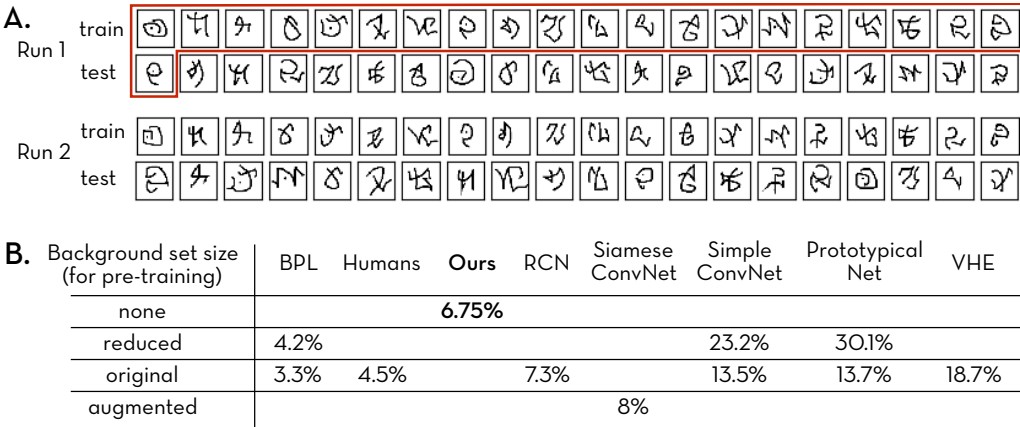

Figure 6: One-shot classification in the Omniglot challenge (A) and its error-rate leaderboard (B). The red bounding area marks one out of 400 unit tasks, made up of 1 test and 10 training images.

**EMNIST-letters in the tiny-data regime.** The original benchmark has 4.8k training images per class and 0.8k test images per class, spanning 26 classes of case-insensitive English letters. We keep the same experimental setting as in MNIST (except for TextCaps being more stable now: we do 7 independent runs for each $N$ and trim the best and the worst). Results are shown in Figure 5. EMNIST-letters is harder, not only with more classes but also more intrinsic ambiguities, e.g., an *l* and an *I* can look identical, so can an *h* and an *n* when written carelessly. Hence, all models perform significantly worse than in MNIST. The intrinsic ambiguity, as well as more labeling errors, narrows our superiority over other models as training size increases. This is especially true for the state-of-the-art TextCaps model, catching up quickly in Figure 5. Being sensitive to ambiguities and outliers, however, is not a deficiency of our distortable canvas model, but a property of nearest-neighbor. To improve, we may integrate our model with more robust classifiers, e.g., $k$-nearest-neighbor ($k$-NN) with proper voting. However, $k$-NN is not applicable in the tiny-data regime, not only because the training size can be as small as $k$ but also there is little room to hold out a validation set for selecting $k$. An adaptive $k$-NN may be desired, with $k$ remaining 1 in the tiny-data regime and becoming tunable when training size increases to a level that affords a held-out validation set. A related issue due to lacking validation data is about picking a proper model configuration. One may expect better results from any selected model in Figures 4 and 5 by attempting new configurations. Yet, it is unclear what heuristics one may use. For TextCaps, we used its original implementation and configuration; for the rest, we used scikit-learn implementations with default configurations (except for small tweaks for the tiny-data regime e.g., neural-network size and stronger regularization). By contrast, our distortable canvas model requires little to tune, other than the $(\hat{G}, \rho_c)$-solution path. Theoretically, the more gradual the path, the better. We picked $(\hat{G}, \rho_c)$ based on only image size (28 here) and runtime.

**The Omniglot challenge for one-shot classification**. The Omniglot dataset contains handwritten characters from 50 different alphabets, which include historical, present, and artificial scripts (e.g., Hebrew, Korean, "Futurama") and thus, are far more complex than MNIST digits and EMNIST letters. The characters are stored as both images and stroke movements. Unlike MNIST/EMNIST coming with large training data, the Omniglot challenge was specially designed for human-level concept learning from small data. Its one-shot classification task was benchmarked to evaluate how humans and machines can learn from a single example. This benchmark contains 20 independent runs of 20-way within-alphabet classifications. The $(2k-1)$th and $(2k)$th runs for $k = 1, \dots, 10$ use the same set of 20 characters from a single alphabet. Each run uses 40 images: one training and one test image per character. The unit task here is to predict for each test image, the character class to which it belongs (one of 20), based on the 20 training images. In total, there are 400 independent unit tasks across all 20 runs. Figure 6A shows a unit task (in red) and the first two runs in the benchmark, covering 1 alphabet, 20 characters, and 80 distinct images.

The Omniglot benchmark adopted the standard FSL setting, where it also provided a background set for pre-training. The original background set contains 964 character classes from 30 alphabets; a reduced background set was proposed later to make the classification task more challenging. We run our $\mathcal{D}_C$-nearest-neighbor without any background set or any stroke-movement information. In

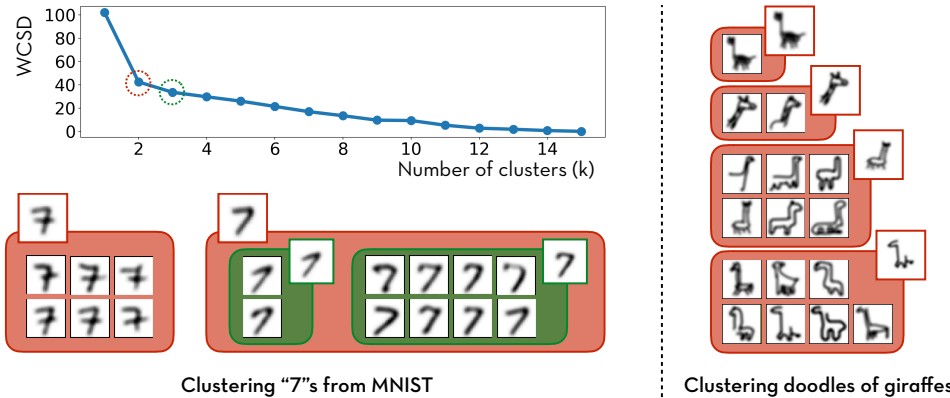

Figure 7: Archetype generation via $k$-means-style clustering in our learned similarity space.

other words, in each unit task, we predict the test image based on *one and only that* training image per character, and we read all images from their raw pixels. Shown in Figure 6B, our model (with a $6.75\%$ error rate) approaches human performance ($4.5\%$) and outperforms all models in the Omniglot leaderboard (Lake et al., 2019), except for BPL specially designed for the Omniglot challenge by making additional use of both the background set and the stroke-movement information.

## 5 UNSUPERVISED LEARNING: ARCHETYPE GENERATION

Beyond use with a classifier, our distortable canvas model can also perform $k$-means-style clustering, but within its human-intuitive topological similarity landscape. As in other metric learning and non-Euclidean settings (Cuturi & Doucet, 2014), a naive way of explicitly computing distances and then running $k$-means is not realistic. Learning distance in our model requires solving an optimization problem, which is not as cheap as computing Euclidean distance. Further, computing a "centroid" in non-Euclidean space requires solving another optimization problem (i.e., minimizing the sum of within-cluster distances)—so an optimization problem of optimization problems—which is not as simple as an arithmetic mean. Our idea of transformation flow between two images can be extended to multi-flows among multiple images. Under this multi-flow extension, we do not explicitly compute pairwise distances, i.e., we do not solve the inner optimizations first. Instead, we solve the inner and outer optimizations at the same time, flattening the nested optimizations into a single one. Formally, given $N$ images $\mathcal{M}_1, \ldots, \mathcal{M}_N$, to cluster them into $K$ clusters, we solve

$$\operatorname*{minimize}_{\substack{\alpha_1,\ldots,\alpha_N \\ \overline{\alpha}_1,\ldots,\overline{\alpha}_K \\ C_1,\ldots,C_K}} \sum_{k=1}^{K} \sum_{i \in C_k} \mathcal{D}_C(\overline{\mathcal{M}}_k \circ \overline{\alpha}_k, \ \mathcal{M}_i \circ \alpha_i) \quad \text{subject to} \sum_{i=1}^{N} \mathcal{D}_V(\alpha_i) \leq \epsilon, \tag{7}$$

where $C_k$ denotes the $k$th cluster, $\overline{\mathcal{M}}_k \circ \overline{\alpha}_k$ denotes the $k$th centroid, and $\mathcal{M}_i \circ \alpha_i$ denotes the $i$th transformed image flowing to its corresponding centroid together with all other $N-1$ transformed images. One can check (7) is an extension of (4) where we omitted $\chi$ for simplicity. Solving (7) is similar to $k$-means via *alternating refinement*: the assignment step assigns each transformed image $\mathcal{M}_i \circ \alpha_i$ to $C_{k^\star}$ according to $k^\star = \arg\min_{k=1,\ldots,K} \mathcal{D}_C(\overline{\mathcal{M}}_k \circ \overline{\alpha}_k, \ \mathcal{M}_i \circ \alpha_i)$; the update step solves (7) for one gradient-descent step given the $C_k$s. Upon convergence, we obtain $C_1^\star, \ldots, C_K^\star$ as clusters and $\overline{\mathcal{M}}_1 \circ \overline{\alpha}_1, \ldots, \overline{\mathcal{M}}_K \circ \overline{\alpha}_K$ as centroids. We treat the learned centroids as *archetypes* of the $N$ images given at the beginning, which may be further deployed for education purposes.

We run the above procedure to generate archetypes of a particular image class. Like common practices used in $k$-means, we try different $k$. For each $k$, we try multiple random starts and record the best within-cluster sum of distances (WCSD). We use the elbow method to pick good values for $k$. Figure 7 shows the WCSD-versus-$k$ curve obtained by running our clustering method on a set of 16 images of "7"s from MNIST. The curve indicates $k = 2$ or 3 as a potential elbow point. The resulting two clusters of "7" agree with human intuition regarding two general ways of writing "7", depending on whether there is an extra stroke. The resulting three clusters further divide the cluster of "simpler 7s" based on the angle of the transverse stroke. Moving away from more strict symbol systems towards

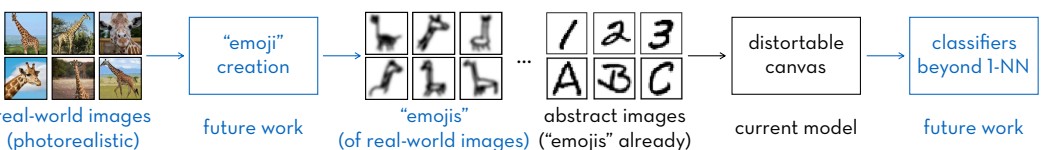

Figure 8: Generalization of our distortable canvas model for two major future directions (two ends).

free-form art, we try our model on doodle data where people were tasked with freely drawing abstract sketches of real-world objects. Figure 7 also shows four ways of doodling a giraffe, learned from the first 16 giraffes in Google's Quick Draw dataset (Jongejan et al., 2016). The four learned archetypes cleanly separate outline sketches, pose orientations, as well as focused views of the neck.

## 6    CONCLUSION, LIMITATION, AND FUTURE WORK

Focusing on its scientific contribution, this paper designs a human-intuitive model from first principles to learn from few and only those few shots—in particular one and only one shot—requiring no extra data for pre-training. Based on nativism, our introduced distortable canvas effectively models humans' topological intuition and learns transformation-based visual similarity akin to how humans naturally "distort" objects for comparison. This notion of similarity is formalized in our proposed optimization problem, which minimizes canvas and color distortions so as to transform one object to another with minimal distortion. To remedy vanishing gradients and solve the optimization efficiently, we mimic human abstraction ability by chaining anchor lattices and image blurs into a solution path. This yields our gradient descent method capable of optimizing at multiple levels of abstraction. Our model outputs not only transformations but also transformation flows that mimic human thought processes. We demonstrate initial empirical success in a first set of benchmarks focused on abstract visual tasks such as character and doodle recognition. By simply using 1-NN, we achieved state-of-the-art results in the tiny-data and single-datum regime on MNIST/EMNIST and achieved near-human performance in the Omniglot challenge. Our model also enables $k$-means-style clustering to generate human-interpretable archetypes. This paper is a first step towards a general theory of a comprehensive, human-like framework for human-level performance in diverse applications. The current paper focuses on an initial scope, but opens the pathway to future generalizations as detailed below.

Consider two general types of images: 1) images of abstract patterns, or *abstract images*, e.g., those of symbols and doodles; 2) photorealistic images of real-world objects, or *real-world images*, e.g., those in CIFAR10/100 (Krizhevsky & Hinton, 2009). This paper focuses on the first type, handling abstract images only, by modeling humans' distortion-based intuition. For real-world images, it may be more efficient to first model cognitive simplification and then apply our current distortion model. It is a reasonable assumption that humans have evolved to classify real-world images by first converting them into abstract icons, or *"e (picture)+moji (character)"*s (e.g., the emoji of a face, the outline of a mountain, the shape of a lake) and then comparing these simplifications. Following this, an efficient way to apply our method to real-world images is to follow this pipeline—preprocessing them first into "emojis" and then comparing "emojis" using our distortion model. In fact, abstract images (e.g., doodles of giraffes, hieroglyphics) are those that can be treated as "emojis" already. There are baselines to attempt first, e.g., smart edge detectors (Xie & Tu, 2015), but the human visual system does more than edge detection. In the future, we will work on a complete theory of icon or "emoji" creation mimicking human capacity and deal with real-world images, 3D objects, and more.

Although our model has shown dominant classification performance in the tiny-data regime of the presented benchmarks, its dominance diminishes when training size increases. This is due to 1-NN being 100% biased towards the "nearest neighbor" and hence fragile against noisy, erroneous, and ambiguous training examples. This suggests another future direction: our distortable canvas model may be designed jointly with a new, human-like classifier that introduces a small amount of learning into classification. The goal is to achieve state-of-the-art results on all training sizes, which is not merely about swapping in and out existing classifiers. We will not be using black-box models, but maintain model interpretability by modeling "the direct human way", where we learn (from a small training set) a particular function to be integrated into our distortion formulas. We will introduce such functions in future work and continue to improve them. Figure 8 summarizes the pipeline for generalizing our current distortable canvas model into the future directions sketched above.

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
