# OpenReview forum: "Learning from One and Only One Shot"
_ICLR.cc/2022/Conference — ICLR 2022 Submitted_

### Official Review · Reviewer_gUCx · 2021-11-02

**Correctness:** 3
**Technical Novelty And Significance:** 3
**Empirical Novelty And Significance:** 3
**Recommendation:** 6
**Confidence:** 5

**Main Review:**

Strengths
1.	This approach is a white-box model and human-interpretable.
2.	This paper is well written and easy to understand.
3.	The major idea is novel and interesting. This is a good attempt to explore a solution for the few-shot learning without pretraining or with very limited training data. Since most existing FSL models heavily rely on a large number of training data.

Weaknesses

1.	The proposed approach is based on topological similarity. It seems that it only is suitable for images with simple topological structure, such as the character images. Maybe it is hardly used to classifier complex nature images since we need more information for natural image classification, not only use topological structure.
2.	The authors did not provide the experimental comparison with enough training data, such as the whole training set in MNIST. The reviewer wonders about the upper performance of this approach with enough data.
3.	The future applications of this approach may be limited, especially for the real-world tasks which require learning from a few labeled data, such as rare accident data in auto-drive and disease data for medical diagnose.


**Summary Of The Paper:**

This paper proposes a novel white-box model for one or few-shot learning, which tries to simulate the human recognition ability for “distort” objects. The authors use transformation-based topological similarity to build the model and propose an anchor system with gradient descent to train their models. Extensive experiments on standard character recognition benchmarks demonstrate the proposed method outperforms the classical machine learning methods with very limited training data, such as less than 20.

**Summary Of The Review:**

The main idea is novel and interesting, and the experimental results on limited training data demonstrate the superior performance of the proposed method. However, it may be not easy to apply this approach to other more complex tasks. In summary, the reviewer thinks this paper is marginally above the acceptance threshold.

---

> ### Author Response · Authors · 2021-11-17
> **Re: Official Review of Paper2362 by Reviewer gUCx (part 2)**
>
> 2. We agree with the Reviewer that although our model has dominant classification performance in the tiny-data regime shown in the presented benchmarks, its dominance diminishes when more training data is available. This is due to the classifier we selected. In this paper, we focused on the tiny-data domain and focused on learning human-intuitive topological similarity (which itself is not a classifier) but spent minimal effort in selecting the classifier. We simply used the vanilla, parameter-free, and hyper-parameter-free nearest neighbor method (1-NN). On the one hand, we use this to show the power of our metric-learning model: even with the simplest classifier in the world, we achieve top results in the tiny-data and single-datum regime. On the other hand, by definition, the nearest neighbor method is well known to be 100% biased towards the “nearest neighbor” and therefore, is least robust against “bad”, noisy, erroneous, or intrinsically ambiguous (e.g., “I” and “l” in some fonts) training examples. As a result, the nearest neighbor method is the underlying reason that slows the progress as the training size increases.
>
> &nbsp;&nbsp;&nbsp;&nbsp;&nbsp;&nbsp;&nbsp;&nbsp;
> The above suggests another future direction: our distortion model will be collectively designed with a new, human-like classifier that introduces a small amount of learning into classification. The goal is to achieve an all-range winner that exhibits state-of-the-art results on all training sizes. This is not about simply swapping in existing classifiers. In particular, we will not be using artificial neural networks. Rather, our plan is to continue to maintain model interpretability and to model “the direct human way” by learning (from a small training set) a particular function to be integrated into the distortion formulas. We will introduce such functions in subsequent papers and continue to improve them.
>
> 3. We thank the Reviewer for raising a very good point on rare accident data in auto-drive and disease data for medical diagnosis. This aligns with our ongoing work on studying the historical evolution of human writing systems in different cultures, e.g., how Brahmi evolved to modern Brahmi-derived scripts such as Bengali, Tamil, Thai, and Tibetan. (Indeed, we believe our algorithmic framework may be able to automatically reproduce the work of James Prinsep, secretary of the Asiatic Society of Bengal in the 1830s, to decode the Brahmi script from a very small number of Brahmi epigraphs and access to modern scripts.)  Besides the “emoji” technique mentioned in Part 1 in handling nature images, the transformational nature of our model makes temporal applications like in auto-drive and medical diagnosis good candidates for our model to experiment with data-scarce problems in time-series data. Besides classifying e.g., danger from safety or disease from health, our distortion model also outputs a transformation flow (i.e., a continuously evolving sequence of transformations), which offers a convenient way to study the evolution of one image or another, e.g., to study how safe situations transform to accidents, or how healthy tissues transform to cancers. Being able to learn not only the end result (e.g., being positive or negative) but also the entire transformation process (e.g., how positive gradually becomes negative) from limited training data, further allows the resulting AI to make preventative warnings in the applications mentioned by the Reviewer.
>
>
> We are completing our revision of the paper according to all Reviewers’ comments. Meanwhile, we welcome any follow-up questions and suggestions in the coming days to help us further improve the paper before its final revision.

---

> > ### Comment · Reviewer_gUCx · 2021-11-28
> > **Response**
> >
> > Thank you for addressing the concerns. After reading the rebuttal of the authors and the comments of other reviewers, the reviewer thinks the main shortcoming is that it is difficult to apply the current model to "real-world images" or more practical applications. There is still a long way to develop this model. However, this is a new attempt to explore the few-shot learning ability of humans, which is very different from the mainstream deep learning models. To encourage a new direction in the community, the reviewer keeps the original rating and attends to accept this paper.

---

> > > ### Author Response · Authors · 2021-11-29
> > > **Re: Response**
> > >
> > > We thank the Reviewer again for appreciating our effort to develop human-like and human-interpretable ML to achieve human-level learning efficiency in our proposed few-shot and only few-shot learning setting.  As Lake, Salakhutdinov, and Tenenbaum (2019) say about handwritten characters: “Characters are far more complex than the low-dimensional artificial stimuli used in classic psychological studies of concept learning, and they are still simple and tractable enough to hope that machines, in the near future, will see most of the structure in the images that people do…. Omniglot is an ideal testbed for studying human and machine learning, and it was released as a challenge to the cognitive science, machine learning, and artificial intelligence (AI) communities.” We are actively building from here!

---

> ### Author Response · Authors · 2021-11-17
> **Re: Official Review of Paper2362 by Reviewer gUCx (part 1)**
>
> We thank the Reviewer for the thoughtful comments! We are glad the Reviewer appreciates our model’s interpretability and our focus on few-shot learning with no pretraining—the two main aspects in our pursuit of human-level performance. The Reviewer is accurate about the scope and capacity of our current model. This paper focused on the **scientific contribution** first (rather than being purely application-centric) with empirical success established in the first set of applications. The current model initiates a line of our ongoing work that collectively aims for a comprehensive framework to achieve human-level concept learning in a variety of applications.
>
> The unaddressed issues raised by the Reviewer are all important points requiring a well-presented theory and implementation rather than a simple algorithmic or experimental twist. A full presentation of each unaddressed point requires a separate paper that we are working on. In this paper, we realize that it is important to mention and sketch these unaddressed issues, so one can see that they are not the limitations of the current model, but rather what can be achieved from the current model. We describe them in detail below.
>
> 1. The Reviewer is correct that our current model focuses on “abstract images” (digits, characters, doodles) instead of “real-world images” i.e., photo-realistic images of real-world objects. Indeed, as the Reviewer pointed out, “we need more information for natural image classification”. This paper is the first step towards eventually handling all images including “real-world images”. Humans memorize only essential features of “real-world images”. It is a reasonable assumption that humans have evolved to classify “real-world images” by first converting them into abstract “emojis” (e.g., the emoji of a face, the outline of a mountain, the shape of a lake) and then comparing those “emojis”. The efficient way to apply our method to “real-world images” is to do it the human way—preprocessing them first into “emojis” and then comparing using our current distortion model. “Abstract images” are in fact those that can be treated as “emojis” already. This is especially apparent from the relation between the doodle image of a giraffe and the photo of a real giraffe, or hieroglyphic symbols abstracted from real-world objects. Due to the page limit, our current model handles “emojis” only; our next paper will introduce the technique for creating them (work in progress). There are some baselines to try first, e.g., smart edge detectors as in [1]. However, the human vision system does more than edge detection. We will work on a complete theory of “emoji” creation in the human way and experiment with “real-world images”, 3D objects, and data beyond computer vision (CV).

---

### Official Review · Reviewer_CPtf · 2021-11-02

**Correctness:** 4
**Technical Novelty And Significance:** 4
**Empirical Novelty And Significance:** 4
**Recommendation:** 5
**Confidence:** 5

**Main Review:**

The paper on hand tackles the problem of data variety in a different way. As  the typical approach nowadays would be to use some kind of data augmentation, here vice-versa some kind of transformation are proposed to provide meaningful similarities. The overall idea is not new, however, the specific way how this is realized is. The overall approach make sense and is backed up by a clear mathematical definition. From this point of view the paper would be a valuable scientific contribution. On the downside, however, there are flaws in the description and interpretation of the results. Moreover, it is unclear if the proposed approach would also work for more realistic datasets: MNIST and EMNIST might be to simple. On the other hand, the Omniglot challenge seems to be a too complex tasks, that no existing approach can solve not even rudimentary. So using this dataset was probably not a good choice?


Further Comments:

(1)  Using partially blue text is hampering the reading flow.

(2) When describing the datasets in Sec.4 the correct abbreviation (given the real numbers) should be 60k and not 60K.

(3) To make the experimental setup more clear, all methods that are used for training should be mentioned explicitly.

(4) For readability, it might be better to split Fig.4 into two figures, one for each benchmark,

(5) Even though the results presented in Fig.4 are clear, a better description of the experimental setup and some kind of discussion would be necessarily required.

(6) For better understanding, it might be meaningful also to show examples for MNIST and EMNIST.

(7) To allow for a fair evaluation and to give better insights, it might be meaningful to use the same learning approaches for all datasets.

(8) The ocker background in Fig.6 does not look nice and removes the focus from the main content of the figure.

(9) The archetype generation does not fully fit to the rest of the paper and is not described and evaluated sufficiently. In this way, it would be better to describe the other experiments more detailed and to shift this part to a separate publication.

(10) In general, the paper is missing a conclusion and an interpretation of the results.

(11) The discussion on related work is rather short, which can also be seen from the very short bibliography. A more thorough discussion would be necessary.

(12) The bibliography needs to be seriously checked for consistency and correctness.

**Summary Of The Paper:**

The paper on hand tackles the problem of learning from very few samples, which is of high relevance for many machine learning problems. To this end, an approach to model transformation-based topological similarity is introduced, allowing for covering many kinds of invariants in image data. The approach is demonstrated for well-known benchmark datasets for different classification models, demonstrating that in this way just using a small number of samples competitive classification results can be obtained.

**Summary Of The Review:**

Overall, the paper comes up with an interesting approach to learn from very few samples. The approach is plausible, backed up by theory and seems to work for simple tasks. However, it seems to be questionable if this would also work for more realistic datasets. Moreover, the experimental section would need to be overworked. In this way, the paper would be an interesting contribution, but there are still a few open points hampering a publication as it is.

---

> ### Author Response · Authors · 2021-11-17
> **Re: Official Review of Paper2362 by Reviewer CPtf**
>
> We appreciate the Reviewer’s positive feedback on our **scientific contribution** as well as the novelty of our specific realization! We thank the Reviewer for all the listed comments to help us improve!
> We agree that our current model is not as comprehensive yet as neural networks for handling a wide range of CV benchmarks and beyond. This paper represents a first step towards a comprehensive framework with desired human-interpretability and near-human efficiency. We realize the current paper needs to be better positioned among not only related work but our other ongoing work that is beyond the current scope. Combining our ongoing work with the current model will lead to our desired comprehensive framework. Below, we clarify the position of the current paper and address all other comments in detail.
>
> **Current scope.** If we divide all images in the world into two types: 1) images of abstract patterns (let’s call them “abstract images”, e.g., those of writing symbols and doodles) and 2) photo-realistic images of real-world objects (let’s call them “real-world images”, e.g., those in CIFAR10/100), this paper focuses on the first type and only considers the task of recognizing “abstract images”. We realize that this was only lightly specified in the abstract and introduction (where we phrased: “abstract visual tasks like character or doodle recognition”); we have made this clear in the revision.
>
> **Extension to wider scopes.** This paper is the first step towards eventually handling all images including “real-world images”. Humans memorize only essential features of “real-world images”. It is a reasonable assumption that humans have evolved to classify “real-world images” by first converting them into abstract “emojis” (e.g., the emoji of a face, the outline of a mountain, the shape of a lake) and then comparing those “emojis”. The efficient way to apply our method to “real-world images” is to do it the human way—preprocessing them first into “emojis” and then comparing using our current distortion model. “Abstract images” are in fact those that can be treated as “emojis” already. This is especially apparent from the relation between the doodle image of a giraffe and the photo of a real giraffe, or hieroglyphic symbols abstracted from real-world objects. Due to the page limit, our current model handles “emojis” only; our next paper will introduce the technique for creating them (work in progress). There are some baselines to try first, e.g., smart edge detectors as in [1]. However, the human vision system does more than edge detection. We will work on a complete theory of “emoji” creation in the human way and experiment with “real-world images”, 3D objects, and data beyond computer vision (CV).
>
> **Omniglot dataset.** We used the MNIST/EMNIST dataset and the Omniglot dataset for different purposes. The former was used in the tiny-data setting to compare our model with other classifiers, whereas the latter was used in its original one-shot setting to compare our model with other few-shot learning (FSL) models as well as humans. The Omniglot Challenge is indeed considered challenging in a sense that the performances of generic state-of-the-art FSL models exhibit large gaps from human performance. Unlike in the experiments with MNIST/EMNIST, the performance of our model—the only model in the Omniglot leaderboard that requires zero pretraining—is not only evaluated based on comparison to other ML models but also based on comparison to human performance. As a result, our conclusion from the Omnilgot Challenge is not only about outperforming other models, but more importantly, about achieving near-human performance by adopting human-like learning strategies.
>
> **Response to the detailed and specific comments.**
>
> (1) We have changed them to black.
>
> (2) Done.
>
> (3) Done.
>
> (4) Done.
>
> (5) Done.
>
> (6) Done.
>
> (7) For MNIST and EMNIST-letters, we used the same set of learning approaches in the comparison. Omniglot Challenge has its own one-shot classification set up, so we directly used its leaderboard for comparison. In case we misunderstood the Reviewer, could you clarify? Thanks!
>
> (8) We have changed the background color to white.
>
> (9) We have shortened it for just illustration purposes, which creates more room for other experiments. In a newly added conclusion and discussion section, we further use this shortened part to connect to our other ongoing work.
>
> (10) Added.
>
> (11) Expanded.
>
> (12) Done.
>
> We are completing our revision of the paper according to all Reviewers’ comments. Meanwhile, we welcome any follow-up questions and suggestions in the coming days to help us further improve the paper before its final revision.

---

### Official Review · Reviewer_7LJy · 2021-11-03

**Correctness:** 3
**Technical Novelty And Significance:** 4
**Empirical Novelty And Significance:** 2
**Recommendation:** 5
**Confidence:** 4

**Main Review:**

**Strong points**: The authors connect ideas in a beautiful way. I liked how they connected ideas from how the human brain works to train their model. Well-structured writing. The idea seems novel and interesting. We optimize steps to generate the target image from our trainset and then pick the one that is closest. I really liked the idea. I also liked making the model work as a white box by using their method. Also finding ways to apply their method in higher levels.

**Weak points**: Biggest drawback is lacking experiments on more challenging datasets, and unfortunately it is very important for me to see the results compared at least on very well-known datasets for few-shot learning.

**Review**
Currently, I gave a score of 5 to the paper.
The main reason for my review is that there are other methods that leverage other data in the domain like (model-agnostic meta-learning[1]) and tested on a few samples on the classes that are never seen during the meta-learning phase. I understand the value of figuring out a way to remove this requirement. For example, unsupervised meta-learning (CACTUs[2]) achieves very good results without needing labelled data. It seems to me that this method has replaced the need for data with a good amount of domain knowledge (We know that characters could convert to each other by transformations, but what about cats and dogs? Another example is identity recognition from faces. If we have two images of two different people. In one picture the person laughs and in the other one, the person is wearing sunglasses. Which two are getting closer to each other?) For example, in figure 1, we know that we can apply these transformations from "7" to "1". However, the experiments did not convince me that this is always true. One interesting experiment that could be added is to apply this method to CelebA[2] face recognition dataset. Another baseline could be to apply this method to the Mini-Imagenet[1] dataset. These are well-known datasets in few-shot learning and studied alongside Omniglot. Results on these could convince the reader about the effectiveness of the method.

**Questions**

In section 2, under "Representative digital and smoothed images" it is mentioned that "This differs from
Gaussian blurring as we do not discretize kernels". Can you specify what you mean? In practice, we must somehow use a discretize kernel on the computer. Can you describe the process in more detail?‌ Do you cut the kernel at some point?

**Other comments to improve the paper**


Related work does not seem comprehensive to me. Only a few papers are cited and some of the statements are kind of misleading. For example, it is true that meta-learning needs data for a pre "meta-learn" step, but there is a nuanced difference. The meta-learning step is applied to different classes or tasks and then evaluated on a completely new task. As a result, I would try to expand this section.

[1] Finn C, Abbeel P, Levine S. Model-agnostic meta-learning for fast adaptation of deep networks. In International Conference on Machine Learning 2017

[2] Hsu K, Levine S, Finn C. Unsupervised Learning via Meta-Learning. In International Conference on Learning Representations 2018



**Summary Of The Paper:**

**Summary**

The authors propose a white-box model that uses a similarity score in a space that is constructed based on topological transformation. In other words, two data points are close to each other if you can transform them into each other. To achieve this, they first compute the topological distance with training data points while minimizing the distortions (this part was a little bit unclear to me. I would appreciate it if the authors provide more context or examples of this). They use an interesting idea of using gradient descent to find a sequence of transformation which is the main reason that contributes to having a white-box model. I found this part of the paper very interesting and novel. Finally, it is not possible to apply this method at the pixel level. So they propose using a chain of lattices to be able to apply their technique in higher abstractions.




**Summary Of The Review:**

As a result, even though I liked the ideas in the paper a lot, I am going to vote for rejection unless I can see the experiments added to the paper.

---

> ### Author Response · Authors · 2021-11-17
> **Re: Official Review of Paper2362 by Reviewer 7LJy (part 2)**
>
> **Empirical comparison to FSL.** As stated above and from the title of this paper, our FSL with zero transfer training and the standard FSL with transfer learning differ in the learning set up. Given the current set up, it is not applicable in many cases to directly compare the two: it may not be fair to forbid the standard FSL from pre-training or meta-learning; whereas it also disadvantages our model at this stage if we allow pre-training while we do not have a pre-training component. As such, the comparison only makes sense in cases where we can demonstrate a clear win: that is, even without pre-training, our model still outperforms FSL with pre-training. This was indeed what we showed in our Omniglot experiment. Our ongoing work aims to further extend our current model to make more empirical comparisons applicable. This includes the “emoji”-creation technique mentioned above aimed for clear wins against FSL in a wider range of vision tasks. We are also working on relaxing our current “few-shot learning, zero-shot meta learning” setting (FSL-0SmL) into a “few-shot learning, few-shot meta learning” (FSL-FSmL) by introducing a small amount of learning but also in “the direct human way”. This will also enable more fair comparisons. Both directions are part of our ongoing work and to be presented in separate papers.
>
> **Clarification on computing the distance by minimizing distortions.** The high-level concept is that two images are considered topologically close, or similar, if very “little effort” (measured by small canvas distortion) is needed to make them look “roughly the same” (measured by small color distortion). We are adding more pictorial examples in the revision to illustrate cases of large and small distortions. The fact that any two images of the same topological type can be transformed into each other is backed by the Riemann mapping theorem (https://en.wikipedia.org/wiki/Riemann_mapping_theorem). In this paper, we have even significantly broadened the scope of the theorem from conformal transformations to any transformations, which makes it possible to transform any image to any other.
>
> **Domain knowledge free.** We adopted the nativist view, so we do not encode any domain knowledge. The illustration of transforming from “7” to “1” is clear. In fact, because we consider all possible transformations, it is always possible to transform dogs to cats, or in general, to transform any image to any other. However, the cost of such transformations (i.e., the distortion we measure) varies. It could be counted as domain knowledge, if one restricts transformations to some special family such as those commonly considered in CV: translation, rotation, scaling, or even affine transformation in general (note: human-intuitive transformation is thought to be much more flexible and does not fall under any of the aforementioned families). As mentioned above, we consider any transformations in this paper, which particularly broadened the scope of Riemann mapping theorem regarding conformal transformations.
>
> **Question on Gaussian blurring.** The standard (discrete) Gaussian blur is done by means of applying a sample Gaussian matrix to each one of the $mn$ pixels in the standard $m\times n$ image grid (i.e., convolution). The matrix is produced by sampling the continuous Gaussian function. We do not sample; and any one of the $mn$ transformed pixels ($x$ in Eq. (2)) can be anywhere in $\mathbb{R}^2$ (not necessarily in the standard grid). The distance ($\rho(z,x)$) between any transformed pixel ($x$) and the center ($z$) of a Gaussian kernel can take any nonnegative real value, and the continuous Gaussian function (i.e., $\kappa$ in the case of Gaussian kernel) is used to compute $\kappa(\rho(z,x))$. There are still finitely many, i.e., $mn$, pixels to be “convolved” but the Gaussian kernel used in the convolution is continuous. Since the locations of transformed pixels are real-valued and keep changing continuously under canvas transformations, we cannot pre-compute a sampling from the continuous Gaussian function. Even more importantly, the continuous Gaussian formula allows us to compute gradients exactly anywhere on $\mathbb{R}^2$.
>
> **In the revision.** Following the Reviewer’s suggestion, we are expanding the related work section and now completing a conclusion and discussion section at the end.
>
> We are completing our revision of the paper according to all Reviewers’ comments. Meanwhile, we welcome any follow-up questions and suggestions in the coming days to help us further improve the paper before its final revision.

---

> > ### Comment · Reviewer_7LJy · 2021-11-27
> > **Response to comments after rebuttal**
> >
> > I thank the authors for their responses to the points in my review. Unfortunately, I was not convinced with the responses from the authors. I will try to explain my reasons here with regards to the different points:
> >
> > **Clarification on computing the distance by minimizing distortions** and **Domain knowledge free**
> > My initial comment was "*It seems to me that this method has replaced the need for data with a good amount of domain knowledge (We know that characters could convert to each other by transformations, but what about cats and dogs?*". What I mean is that consider four images: 1- A cat looking at the camera from a very close distance. 2- A dog looking at the camera from very close distance. 3- A cat very far from camera next to a car. 4- A dog very far from camera next to a car.
> > My concern is that I am not sure that the image 1 is closer to image 3 or image 2 based on only transformations. This means we need domain knowledge instead of data. This does not have anything to do with the fact that any two images of the same topological type can be transformed. My concern is that how do you know which one is closer to the other one.
> > This also shows why your method gets a clear win in Omniglot experiments because the transformations make sense based on domain knowledge. I do not expect to see your method outperforms other methods in other domains, but I hoped I see some results on applying your method on other domains to make sure that the paper conveys the effectiveness of the model. As a result, I keep the same score.
> >
> > **Current scope**
> > I see your point. Even on this scope, I think you need to add more experiments. For example, what if I want to classify characters based on the number of holes in them. For example Capital A has one hole, and B has two holes. Is your method going to work on this task? Is it able to extract features that are useful to handle this or do we need to somehow insert knowledge about the domain to define a new kind of distance? If you focus on this scope, I still am not convinced that this method can generally solve any classification task in this scope.
> >
> > **Question on Gaussian blurring** I see what you mean here. Thanks for the explanation!

---

> > > ### Author Response · Authors · 2021-11-29
> > > **Re: Response to comments after rebuttal**
> > >
> > > We thank the Reviewer for the further discussion! We constructed and ran our algorithm on the specific setting suggested by the Reviewer. We give the result below and clarify potential misunderstandings.
> > >
> > > Our learned topological similarity represents **visual-appearance similarity**—in terms of how any two arbitrary images look in general—rather than **label similarity**. Two images that “look similar” can be labeled differently, e.g., ‘h’ looks closer to ‘n’ than to ‘H’, but EMNIST labels ‘h’ and ‘H’ the same while labeling ‘n’ as a different letter. **We do not assume our learned similarity implies label similarity** (in fact they do not have to match). That we do not require our learned similarity to match label similarity might be the cause for confusion in terms of what we do and why we do not bring in domain knowledge.
> > >
> > > **What our model does.** Our learned topological similarity captures human innate intuition about shape and distortion, e.g., quantifying our intuition about ‘h’ being topologically closer to ‘n’ than to ‘H’. This visually intuitive similarity has no relation to labels. After all, a labeler can label whatever class it wants for the images regardless of whether they “look similar”.
> > >
> > > **What our model does not do.** Our distortion model does not predict label similarity. If a model had to guess/assume the underlying labeling mechanism, it may very likely use domain knowledge; however, our model does not. No label information is ever used in our metric learning process, and our distortion model itself is not a classifier. When our learned similarity is later fed into a classifier for classification, it is that adopted classifier—in our case nearest-neighbor—that learns label similarity, often with several “attraction regions” in the metric space rather than a single connected cluster.
> > >
> > > We run our algorithm on 4 images described by the Reviewer:
> > >
> > >  * img_1: A cat looking at the camera from a very close distance
> > >  * img_2: A dog looking at the camera from very close distance
> > >  * img_3: A cat very far from camera next to a car
> > >  * img_4: A dog very far from camera next to a car
> > >
> > > The cat, dog, and car are extracted from doodle images from Google’s QuickDraw dataset: https://drive.google.com/file/d/1RZDpmqs8e2iCG0DaUd-RQm4Y1fOmwsfQ/view?usp=sharing. The distances returned by our distortion model are:
> > >
> > >  * Distance(img_1, img_2) = 4.4
> > >  * Distance(img_1, img_3) = 5.1
> > >  * Distance(img_4, img_3) = 0.6
> > >
> > > That is, img_1 is visually closer to img_2 than to img_3. This agrees with human intuition that img_1 and img_2 look similar (perhaps because both are face-like shapes) whereas img_3 looks generally different. Our distortion model disregards how the images are labeled. One may certainly label img_1 and img_3 as “cat” and img_2 and img_4 as “dog”. If one wants to predict img_3 (as a test image) correctly in this case, one needs both a classifier and training images that look similar (in our topological sense) to img_3.
> > >
> > > Predicting a visually distinct test image that has never appeared in the training set is challenging for all ML models (this violates the basic statistical assumption that training and test samples are drawn from the same distribution, e.g., predicting an “A” as “a” while no upper-case “A” is ever present in the training set). We create an additional example regarding classification at
> > >
> > > https://drive.google.com/file/d/1Jv2-F1CuccqbTBtymR7scb39ChkXcxad/view?usp=sharing
> > >
> > > An image of a “cat” can certainly look more similar to a “dog”, e.g., train_1 (labeled “cat”) is closer to train_2 (labeled “dog”) than to train_3 (labeled “cat”). A test image (test) can still be predicted correctly as long as some similar looking image (train_3 in this case) is present in the training set. This example emphasizes the point that our metric learning phase and the classification phase are separate, and that the labels only come into play in the classification phase. Note that if the vision task is to predict whether an image contains a cat/dog (e.g., next to a car), it becomes an object detection/segmentation task rather than a pure recognition task. This paper only considers pure recognition.
> > >
> > > Regarding another setting described by the Reviewer, leveraging our learned similarity in the nearest-neighbor method can indeed classify “A” from “B” with high probability (evident from our EMNIST experiment) but we do not compare them based on the number of holes. We use distortion instead, which captures the intricate visual difference between the bottom half of “A” and that of “B”, i.e., how far away the bottom half of “A” is from a hole like the bottom half of “B”.
> > >
> > > In short, saying that two images “look similar” reflects human innate intuition whereas “labeling them the same” may reflect domain knowledge and can be artificial or even random. We model human innate intuition, whereas we do not use domain knowledge involved in classification—our distortion model is not a classifier.

---

> > > > ### Comment · Reviewer_7LJy · 2021-11-29
> > > > **Thank you for your response**
> > > >
> > > > I see what authors mean by "*We do not assume our learned similarity implies label similarity*". As I mentioned the idea is interesting and novel, but there are more experiments that could show the impact and effectiveness of the model. I suggest the authors add these to the paper. For example, how bad does the model work on a real-world image dataset like Mini-Imagenet that is used to report accuracy for many few-shot learning papers.  Furthermore, I am not convinced that this similarity based on transformation is good for any classification task. As I said, if the classification task is to find the number of holes in a character, how this similarity metric impacts the classifier? One good experiment that could be added is as follows:
> > > > 1- Label all English letters based on the number of holes
> > > > 2- Sample one image from each category
> > > > 3- Use your method to do the classification and compare with other methods
> > > > The proposed method is not going to work well, because for example, it might think that C is close to O while for this classification task, we need other kind of features.
> > > > I again must mention that the idea and exploration of this idea is a great direction for work, and I appreciate the authors for working on this. However, it still needs more experiments to convey to the reader a correct picture of the impact of this paper.

---

> > > > > ### Author Response · Authors · 2021-12-01
> > > > > **Re: Thank you for your response**
> > > > >
> > > > > We thank the Reviewer for considering our idea novel and appreciating our current exploration! We are grateful for all the suggested experiments, which tremendously helps improve our current work to better show its impact and effectiveness. Regarding potential doubts on characterizing “holes” and particularly classifying ‘C’ and ‘O’, we explicate more details from our EMNIST-letters experiment. They show our model is efficient for characterizing holes. We present them below.
> > > > >
> > > > > 1. We share the same intuition with the Reviewer about ‘C’ looking similar to ‘O’, and we believe this reflects human intuition in general. Our model accurately captures this intuition, which is exactly what we desire. By plotting the distances of the 26 letters (the first test image per class) to ‘O’ (the first training image from the ‘O’-class), our model reports that ‘C’ is the 3rd closest to ‘O’ among all 25 non-‘O’ letters.
> > > > >
> > > > > 2. However, considering ‘C’ similar to ‘O’ does *not* mean our method does not work well for classifying ‘C’ from ‘O’. This is because in general, ‘C’ is even closer to ‘C’; ‘O’ is even closer to ‘O’.
> > > > >
> > > > > - If we have clearly-written ‘C’ and ‘O’ in the training set, then classification (for letters) will be accurate.
> > > > > - In case we have a ‘C’ carelessly written and form a hole, our model will effectively recognize this hole (even if it is carelessly formed). Because of this, the nearest-neighbor method may mistakenly predict ‘O’s as this O-like ‘C’ (but can recover from the mistake by k-NN and/or more data). Nevertheless, predicting holes rather than letters tends to be correct. This is because without domain knowledge and based purely on the image, this carelessly written ‘C’, as well as most ‘O’s, is more likely to be labeled as “1 hole”. This shows our model is efficient for capturing topological features including holes but also many others.
> > > > >
> > > > > Both points above are evident in a more detailed view of our EMNIST-Letters experiment: https://drive.google.com/file/d/1bOIIyoQ2Su1jNlGFB5eWvgsoQ5iHGjXO/view?usp=sharing
> > > > >
> > > > > - In the above figure, in addition to our currently reported result in terms of the overall test set error (dotted black), we further plotted the error curves for class-‘O’ (dashed orange) and class-‘C’ (dashed blue) individually. Both the ‘O’-curve and ‘C’-curve are below the overall error curve, meaning ‘O’ and ‘C’ are even more accurately classified compared to the state-of-the-art overall classification for all letters.
> > > > > - We further plotted how many ‘O’s (in terms of percentage) are particularly predicted as ‘C’ (solid green) and how many ‘C’s are predicted as ‘O’ (solid red). Such mistakes are even rarer (e.g., from the red curve, only 4% of ‘C’s are falsely predicted as ‘O’ after being trained on just 1 example per class; it drops to 0.625% with 2 training examples per class). Interestingly, we observed that when predicting ‘O’s from N=3 training examples per class (third green dot), there is a rise in confusing ‘O’s with ‘C’. We then plotted the third training image of ‘C’, which is indeed a carelessly written ‘C’ that almost forms a hole—the hole is actually closed if one considers all non-white pixels. This verifies that our model is efficient for characterizing holes even without hardcoding such knowledge into the model.
> > > > >
> > > > > 3. We found the Reviewer’s suggested classification task of “finding the number of holes in a character” interesting in general. Creating such a benchmark requires much manual labor—crowd labeling each image by eyeballing the number of holes in it. We are particularly grateful if the Reviewer happens to know and can already direct us to any existing benchmarks for characterizing holes or Betti numbers in general.

---

> > > > > > ### Comment · Reviewer_7LJy · 2021-12-02
> > > > > > **Response to authors message**
> > > > > >
> > > > > > I am glad that you found these experiments interesting. I see your point that this is impossible to learn to classify O and C from each other with just 1 example because this could be an ill-defined task. One at least needs to see more examples of the task to be able to recognize what are the key features. I would like to increase my score based on the discussion, but I leave this decision to area chairs. Things that should be considered is that this paper only focuses on domain of handwritten characters and sketches. Also, the experimental section needs to be improved based on the discussions to show the effectiveness of the method. I suggest the authors to think about experiments that can validate their focus, and I am not sure if it is possible to be done for the camera-ready version. If the area chair thinks that these could be added to the paper by camera-ready version, you can consider my score as a 6.

---

> > > > > > > ### Author Response · Authors · 2021-12-04
> > > > > > > **Re: Response to authors message**
> > > > > > >
> > > > > > > We are very grateful for these discussions with the Reviewer. All suggestions and advice therein have been inspiring and greatly helped us improve the experiments and better present the effectiveness of our mathematical model. We are finalizing another substantial pass of revision of the paper based on insights from these follow-up discussions.

---

> ### Author Response · Authors · 2021-11-17
> **Re: Official Review of Paper2362 by Reviewer 7LJy (part 1)**
>
> We thank the Reviewer for the thoughtful comments! We are glad the Reviewer appreciates our focused **scientific contribution** on human-style and white-box learning. We agree that our current model is not as comprehensive yet as neural networks for handling a wide range of CV benchmarks and beyond. This paper represents a first step towards a comprehensive framework with desired human-interpretability and near-human efficiency. We realize the current paper needs to be better positioned among not only related work but our other ongoing work that is beyond the current scope. Combining our ongoing work with the current model will lead to our desired comprehensive framework. Below, we clarify the position of the current paper, elaborate on comparisons to few-shot learning (FSL), and address all other questions in detail.
>
> **Current scope.** If we divide all images in the world into two types: 1) images of abstract patterns (let’s call them “abstract images”, e.g., those of writing symbols and doodles) and 2) photo-realistic images of real-world objects (let’s call them “real-world images”, e.g., those in CIFAR10/100), this paper focuses on the first type and only considers the task of recognizing “abstract images”. We realize that this was only lightly specified in the abstract and introduction (where we phrased: “abstract visual tasks like character or doodle recognition”); we have made this clear in the revision.
>
> **Extension to wider scopes.** This paper is the first step towards eventually handling all images including “real-world images”. Humans memorize only essential features of “real-world images”. It is a reasonable assumption that humans have evolved to classify “real-world images” by first converting them into abstract “emojis” (e.g., the emoji of a face, the outline of a mountain, the shape of a lake) and then comparing those “emojis”. The efficient way to apply our method to “real-world images” is to do it the human way—preprocessing them first into “emojis” and then comparing using our current distortion model. “Abstract images” are in fact those that can be treated as “emojis” already. This is especially apparent from the relation between the doodle image of a giraffe and the photo of a real giraffe, or hieroglyphic symbols abstracted from real-world objects. Due to the page limit, our current model handles “emojis” only; our next paper will introduce the technique for creating them (work in progress). There are some baselines to try first, e.g., smart edge detectors as in [1]. However, the human vision system does more than edge detection. We will work on a complete theory of “emoji” creation in the human way and experiment with “real-world images”, 3D objects, and data beyond computer vision (CV).
>
> **Comparison to Few-Shot Learning (FSL).** For many applications, FSL via transfer/meta learning achieves impressive success in data-scarce scenarios. We understand the nuanced difference mentioned by the Reviewer that “the meta-learning step is applied to different classes or tasks and then evaluated on a completely new task”. There are known challenges for FSL too—both from a scientific and a practitioner’s perspective—which are exactly what motivated our current model that requires zero pre-/meta-learning. It is important, as a scientific question, to study how to design machine learning to approach human-level learning efficiency—requiring not only little training but also little pretraining [3]. This aligns with the main theme in Chollet's work [4] and the Omniglot Challenge [2], studying how AI may “learn so much from so little” as humans do. The scientific push for “small transfer” avoids the “black art” of FSL consistently encountered by practitioners. Transfer learning relies on the assumption that source and target tasks are similar enough for the first round of training to be relevant [5]. However, it is a “black art” to characterize a priori how relevant the tasks are. In addition, transfer learning can suffer from negative transfer [6, 7]. For widely studied areas such as CV and NLP, there are standard pre-trained or meta-learned models for generic uses. Yet in special cases, e.g., learning from ancient scrolls [8], and learning from limited user behavioral data collected from an early-stage and brand new business, one lacks background data that is sufficiently relevant to the target data for pretraining.

---

### Official Review · Reviewer_MBhM · 2021-11-04

**Correctness:** 4
**Technical Novelty And Significance:** 3
**Empirical Novelty And Significance:** 1
**Recommendation:** 5
**Confidence:** 3

**Main Review:**

Pros:
1. The proposed model is interesting and neat.
2. The empirical results look promising. With very limited data and no pretraining, the proposed method archives significant results on MNIST, EMNIST and Omniglot.

Cons:
1. The benchmark seems too toy-size. Even the most challenging Omniglot dataset has a clean background and somewhat simple visual appearance. In this sense, I am not convinced whether the method can still work well when the image becomes more complex. When more complicated distortion exists (e.g.), is the model still able to capture it through the transformation? I would expect at least some more complicated benchmark datasets to be evaluated, for example, CIFAR10/100.
2. When the task is more challenging (e.g. EMNIST), the proposed method is caught up quickly by the TextCaps. On more complex datasets, modern deep learning models may catch up earlier, and with acceptable data size, even reach much better results that the proposed method cannot obtain.
3. I am not very convinced about the setting. It has been studied that good pretraining can benefit the downstream tasks even in a few-shot setting [R1]. So, I am not sure when the proposed method can be applied. Can authors give me some examples in real world about training a classifier with very few training samples while no additional pretraining data is available?
3. The paper lacks a conclusion.

Reference:

R1: Kolesnikov, Alexander, et al. "Big transfer (bit): General visual representation learning." Computer Vision–ECCV 2020: 16th European Conference, Glasgow, UK, August 23–28, 2020, Proceedings, Part V 16. Springer International Publishing, 2020.

**Summary Of The Paper:**

This paper proposes a method that models the visual difference between images as a topological transformation for images. Using this model and computing the topological distance by minimizing the distortions between imagers, one can find neighbors that are conceptually similar to the input image. Evaluation results show that this simple method can achieve strong performance when only very few samples are available and no pretraining is allowed.

**Summary Of The Review:**

This paper is interesting and has some promising results. But I have concern about its application in the real world and its robustness in more complex scenarios, which I hope the authors can address. Therefore, I incline to reject this paper.

---

> ### Author Response · Authors · 2021-11-17
> **Re: Official Review of Paper2362 by Reviewer MBhM (part 2)**
>
> 2. We agree with the Reviewer that although our model has dominant classification performance in the tiny-data regime shown in the presented benchmarks, its dominance diminishes when more training data is available. This is due to the classifier we selected. We focused on learning human-intuitive topological similarity (which itself is not a classifier) but spent minimal effort in selecting the classifier. We simply used the vanilla, parameter-free, and hyper-parameter-free nearest neighbor method (1-NN). On the one hand, we use this to show the power of our metric-learning model: even with the simplest classifier in the world, we achieve top results in the tiny-data and single-datum regime. On the other hand, by definition, the nearest neighbor method is well known to be 100% biased towards the “nearest neighbor” and therefore, is least robust against “bad”, noisy, erroneous, or intrinsically ambiguous (e.g., “I” and “l” in some fonts) training examples. As a result, the nearest neighbor method is the underlying reason that slows the progress as the training size increases.
>
> &nbsp;&nbsp;&nbsp;&nbsp;&nbsp;&nbsp;&nbsp;&nbsp;
> The above suggests another future direction: our distortion model will be collectively designed with a new, human-like classifier that introduces a small amount of learning into classification. This is not about simply swapping in existing classifiers. In particular, we will not be using artificial neural networks. Rather, our plan is to continue to maintain model interpretability and to model “the direct human way” by learning (from a small training set) a particular function to be integrated into the distortion formulas. We will introduce such functions in subsequent papers and continue to improve them.
>
> 3. We thank the Reviewer for pointing out that the general setting of this paper and its underlying research problem were not well motivated or explained. We have now made them clear in the revision, which we are completing. Instead of being application-centric and demonstrating that we can nail some applications using much less data, we emphasize that this paper is focused on the science of learning.
>
> &nbsp;&nbsp;&nbsp;&nbsp;&nbsp;&nbsp;&nbsp;&nbsp;
> For many applications we agree that few-shot learning (FSL) via transfer learning achieves impressive success in data-scarce scenarios. Yet, from a scientific point of view, it is also important to study how to design machine learning to approach human-level learning efficiency—requiring not only little training but also little pretraining [3]. The motivation here aligns with the main theme in Chollet's work [4] and the Omniglot Challenge [2], studying how AI may “learn so much from so little” as humans do. Our contribution here is to establish a human-like model and its initial success in the first few benchmarks. Notably, complementary to the “Big transfer” paper cited by the Reviewer, the Omniglot Challenge urges “small transfer”— using reduced background data in one-shot learning’s pretraining. We followed this scientific perspective from the Omniglot Challenge and pushed the background-data reduction to the limit—zero pretraining—while still achieving near-human performance. We hope the Reviewer will consider this viewpoint in reevaluation of the paper revision, with any advice to help us improve in this direction.
>
> &nbsp;&nbsp;&nbsp;&nbsp;&nbsp;&nbsp;&nbsp;&nbsp;
> The scientific perspective we take also avoids the “black art” in FSL. Transfer learning relies on the assumption that source and target tasks are similar enough for the first round of training to be relevant [5]. However, it is a “black art” to characterize a priori how relevant the tasks are. In addition, transfer learning may also suffer from negative transfer [6, 7]. For widely studied areas such as CV and NLP, there are standard pretrained models for generic uses. Yet in special cases (as queried by the Reviewer), e.g., learning from ancient scrolls [8] and learning from limited user behavioral data collected from an early-stage and brand new business, one lacks background data that is sufficiently relevant to target data for pretraining.
>
> 4. We add a conclusion section to the paper at the end.
>
> We are completing our revision of the paper according to all Reviewers’ comments. Meanwhile, we welcome any follow-up questions and suggestions in the coming days to help us further improve the paper before its final revision.

---

> ### Author Response · Authors · 2021-11-17
> **Re: Official Review of Paper2362 by Reviewer MBhM (part 1)**
>
> We thank the Reviewer for the thoughtful comments! We are glad the Reviewer found our model neat and our result promising. As the Reviewer pointed out, we lacked a conclusion summarizing a clear goal and scope of the current model and how they link to our other ongoing research beyond the current scope.
> Focusing on a **scientific contribution** first (rather than being purely application-centric), this paper aims to initiate a line of research that advances human-like AI. We include a conclusion with a clear positioning of the paper in the revision, which we are completing. Below, we address each of the comments in detail.
>
> 1. We agree with the Reviewer that our current model is not comprehensive yet for handling all visual datasets. If we divide all images in the world into two types: 1) images of abstract patterns (let’s call them “abstract images”, e.g., those of writing symbols and doodles) and 2) photo-realistic images of real-world objects (let’s call them “real-world images”, e.g., those in CIFAR10/100), this paper focuses on the first type and only considers the task of recognizing “abstract images”. We realize that this was only lightly specified in the abstract and introduction (where we phrased: “abstract visual tasks like character or doodle recognition”); we have made this clear in the revision.
>
> &nbsp;&nbsp;&nbsp;&nbsp;&nbsp;&nbsp;&nbsp;&nbsp;
> This paper is the first step towards eventually handling all images including “real-world images”. Humans memorize only essential features of “real-world images”. It is a reasonable assumption that humans have evolved to classify “real-world images” by first converting them into abstract “emojis” (e.g., the emoji of a face, the outline of a mountain, the shape of a lake) and then comparing those “emojis”. The efficient way to apply our method to “real-world images” is to do it the human way—preprocessing them first into “emojis” and then comparing using our current distortion model. “Abstract images” are in fact those that can be treated as “emojis” already. This is especially apparent from the relation between the doodle image of a giraffe and the photo of a real giraffe, or hieroglyphic symbols abstracted from real-world objects. Due to the page limit, our current model handles “emojis” only; our next paper will introduce the technique for creating them (work in progress). There are some baselines to try first, e.g., smart edge detectors as in [1]. However, the human vision system does more than edge detection. We will work on a complete theory of “emoji” creation in the human way and experiment with “real-world images”, 3D objects, and data beyond computer vision (CV).
>
> &nbsp;&nbsp;&nbsp;&nbsp;&nbsp;&nbsp;&nbsp;&nbsp;
> The category of “abstract images” has its own complexity. Human-invented symbol systems can be both intrinsically ambiguous (labeling visually identical elements differently: “I” and “l” in some fonts) and artificial (labeling visually distinct elements the same: “A” and “a”). In particular, the Omniglot dataset was especially designed to reveal the large gap between human-level concept learning and modern ML. It is considered challenging for modern ML, which is the main point in the Omniglot papers [2,3] and also mentioned by Reviewer CPtf. Using state-of-the-art (SOTA) performance as the metric, Omniglot Challenge is more challenging than CIFAR-10 (solved nearly perfectly). Our model outperforms all one-shot learning models listed in Omniglot leaderboard and significantly closes the gap towards human performance. The follow-up Omniglot Challenge [3] increased the challenge by further requesting pre-training reduction, while our method works at the extreme with absolutely zero pretraining.

---

### Author Response · Authors · 2021-11-17
**References used in all posts**

[1] Saining Xie, Zhuowen Tu. Holistically-nested edge detection.  in Proc. ICCV, pp. 1395–1403, 2015.

[2] Brenden M Lake, Ruslan Salakhutdinov, and Joshua B Tenenbaum. Human-level concept learning through probabilistic program induction. Science, 350(6266):1332–1338, 2015.

[3] Brenden M. Lake, Ruslan Salakhutdinov, and Joshua B. Tenenbaum. The Omniglot challenge: a 3-year progress report. Current Opinion in Behavioral Sciences, 29:97–104, 2019.

[4] François Chollet. On the measure of intelligence. arXiv preprint:1911.01547, 2019.

[5] Marla Rosner. Transfer learning & machine learning: how it works, what it’s used for, and where it’s taking us. https://www.sparkcognition.com/transfer-learning-machine-learning/.

[6] Sinno Jialin Pan and Qiang Yang. A survey on transfer learning. IEEE Trans. Knowl. Data Eng., 22 (10):1345–1359, 2009.

[7] Amiel Meiseles, Lior Rokach. Source model selection for deep learning in the time series domain. IEEE Access,8:6190–6200, 2020.

[8] Mladen Popović, Maruf A. Dhali, Lambert Schomaker. Artificial intelligence based writer identification generates new evidence for the unknown scribes of the Dead Sea Scrolls exemplified by the Great Isaiah Scroll (1QIsaa). PloS ONE. 16(4):e0249769, 2021.

---

### Author Response · Authors · 2021-11-23
**Full revision uploaded**

We thank all the reviewers for all their thoughtful comments and efforts towards improving our manuscript! We have submitted our final revision by the end of this discussion period.

---

### Decision · Program_Chairs · 2022-01-20

**Decision:**

Reject

**Comment:**

Meta Review of Learning from One and Only One Shot

The motivation of this work is to address the problem of learning from very few samples, which is of high relevance for many machine learning problems. The paper proposes an (interpretable) approach for one or few-shot learning, which tries to simulate the human recognition ability for “distort” objects. To achieve few-shot learning, they first model the topological distance with training data points while minimizing the distortions, to find neighbors that are conceptually similar to the input image. Their experimental results show that this simple method can achieve good performance when only very few samples are available and no pre-training is allowed.

All reviewers, including myself, agree that this paper is well motivated, nicely written, and appreciated how they connected ideas from neuro-psychology to their ML model, and the novelty is recognized. But there are issues raised by reviewers with the paper that prevent it from meeting the bar for me to recommend it for acceptance at ICLR 2022.

The main issues raised by all reviewers is that the proposed method is only experimentally verified on simple datasets such as MNIST, EMNIST and Omniglot (and to some extent, Quick, Draw!). The authors (to their credit) in the rebuttal noted that the narrative of the paper is to focus on abstract images, and the purpose is more from a scientific investigation perspective (rather than proposing an algorithm that is immediately useful for ML practitioners), and this is a fair point. However, I do believe the issue here is beyond the simple criticism of "it works on MNIST, how about ImageNet?" as I think some reviewers genuinely think there are fundamental aspects of the approach that might prevent it from scaling (as planned in future work, even by the authors in the last section). For instance, as gUCx noted:

1) The proposed approach is based on topological similarity. It seems that it only is suitable for images with simple topological structure, such as the character images. Maybe it is hardly used to classifier complex nature images since we need more information for natural image classification, not only use topological structure.

2) The authors did not provide the experimental comparison with enough training data, such as the whole training set in MNIST. The reviewer wonders about the upper performance of this approach with enough data.

I tend to agree with these points. Non-topological similarity can be displayed in abstract images / datasets. Even in "abstract" images, the paper should describe limitations of the approach, and whether it breaks down (like in the "abstract" Quick, Draw! dataset, there are different types of distinct "yoga" poses in the yoga class. And likewise, in the cat or pig class, there are animals with only the heads, and animals with the head and the full body). Conveniently, Quick, Draw! had not been used in any of their classification experiments [1], and only for a simple clustering example.

And for the other points, reporting the terminal MNIST performance will be useful, even if it doesn't look good, so the readers have an idea of the limitations of the approach, where it is good, where it is not, and what needs to be improved. I would love to see improvements (either in the writing or in the experiments) in future work where the paper can effectively convince the readers that the direction has the promise of being able to scale to "real" or "complex" images. (Perhaps even performing the approach on the output of a pre-trained self-supervised autoencoder on ImageNet, as a method to get "abstract" versions of real photos, like a parallel of the giraffe experiment, though this may distract from the narrative of no-pre-training). All in all, I don't want to discourage the authors as we are all excited about the direction of this work. I hope to see an updated version of this work published in a future venue, good luck!

[1] https://www.kaggle.com/c/quickdraw-doodle-recognition